# Fiscal Measurement and Oil and Gas Production Market: Increasing Reliability Using Blockchain Technology

Carlos Barateiro [1,*], Alexandre Faria [2], Jose Farias Filho [2], Karolina Maggessi [2] and Claudio Makarovsky [2]

1  Department of Engineering, Campus Macae, Estácio de Sá University, Macaé 28895-270, Brazil
2  Department of Engineering, Campus Niterói, Fluminense Federal University, Niterói 28895-270, Brazil
*  Correspondence: cerbb@terra.com.br; Tel.: +55-(22)996158101

**Featured Application: Use of blockchain technology in the oil and gas market.**

**Abstract:** The market of oil and gas has many particularities, once it is a natural resource of great value. Due to the fact the resource is owned by nations, it is highly regulated. The volumes traded are high; for this reason, their calculation needs to be performed very carefully, meeting not only the uncertainties and metrological control, but also particularly the tracking of the activities. The technical regulations of measurement adopted by the countries carefully follows many guidelines. The reason is that the established volumes directly affect the calculation of royalties and profit sharing in concession agreements or even a simple ownership shift of the products. Therefore, it is an application with a lot of responsibility, involving a large amount of equipment, software, and execution processes. Therefore, the transfer of data among different entities requires total transparency and security. Blockchain technology, which has been initially developed for the financial market, presents itself as an alternative to ensure reliability, from the sensors in the field to the effective generation of the Monthly Report on the petroleum and natural gas production unit, which is the basic document for determining the remuneration of the owners of the product. This paper presents a technical solution for creating the blockchain validation blocks by the MAC (Media Access Control Address) addressing, which in turn comes from the communication boards of the flow computers and from the Supervisory Stations. There are limitations to extending this solution to the level of field sensors due to the current links of communication, but also because of the way that historical, events and alarm databases of the flow computers are generated. Once these devices exhibit an elevated degree of safety in their operation, the solution herein presented adds a high level of reliability in the fiscal measurement and/or custody transfer.

**Keywords:** fiscal measurement; flow measurement; blockchain technology

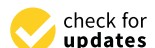

## 1. Introduction

The industry of oil and gas is a highly regulated market because it involves many stakeholders, once it handles and transfers large volumes of high-value products [1]. Crude oil and natural gas are some of the most traded commodities globally, being the feedstock for fuel products, such as gasoline, diesel, and many other petrochemical derivatives. This market, with special attention to the segment of exploration and production, has been looking for new solutions, focusing mostly on the digitization of the operations. However, instead of innovating, most initiatives still seek only to improve something that already exists. Additionally, the resistance to change can be explained precisely either by the excess of standards and regulations or by the various safety aspects involved in the activities of the production units [2]. Another similarly important factor about this market is that it is common to come across outdated technologies under the control of the producers, once their offshore platforms and onshore units were installed many decades ago, consequently making it more challenging and costly to upgrade them [3]. However,

there is an application that has been attracting extraordinary interest in the market: fiscal measurement and/or measurement through the transfer of ownership.

Typically, natural resources are under the ownership of nations. Therefore, there is a need for specific authorizations (concessions) and the establishment of rules for the remuneration of the real owners, the society. Over the decades, such models have been through development, giving rise to three basics, internationally adopted types: concession, sharing, and mixed regimes. They differ mainly in the level of intervention of the state and the distribution of the oil revenues. In general, OECD countries prefer to adopt the regime of concession, while Middle Eastern countries prefer the sharing model. The volume of reserves, the consumption of each country, and the production structure affect the preference for a particular kind or another. In the concession model, the government carries out auctions of exploration sites for which companies compete, paying taxes or royalties over the earnings resulting from the oil production. In the share contract model, those who win the auction pay a flat-rate subscription bonus, and the government is entitled to a percentage of the oil and/or gas produced, which is called "Profit oil" or "Profit gas" [4].

Both the concession and sharing models stem from the same central idea: the reliability in the process of quantification of the volume of oil produced. This quantification makes it possible to define how much the concessionaire should pay in royalties and taxes. It also allows the determination of the percentage held by the government in the sharing. For this reason, this quantification is extremely regulated. Plant operators need to ensure low levels of uncertainty, traceability of actions, and high quality of measurements. They must consider technical measurement regulations specified by the regulatory agents of the operating contracts and count on the direct participation of agents responsible for the metrological aspects of each country [5].

As much as these regulations seek to create such conditions, there is still room to improve transparency, efficiency, and optimization concerning data flow. It covers solutions from sensors and field meters to flow computers (that effectively run the algorithms to calculate the standard volumes) and the monitoring stations (interfaces with operators) until they reach the production report, which in turn reports to the regulator the volumes produced, consumed, and offloaded into the reservoirs. Throughout this data transfer process, failures may occur. In most cases, however, it is precisely the last stage that is the most fragile and subjected to manipulation and errors, once this step is performed manually. Thus, the objective of this work is exactly to propose a solution based on blockchain technology and based on the concepts of the Internet of Things (IoT), which ensures the security of data transferred from the field (sensors and meters) to the measurement bulletins (final stage). They become dynamic, fully automated, independent from human intervention-subjected operations, and highly reliable.

Blockchain is a distributed ratio technology that incorporates a decentralized network, where each transaction is immutably recorded, verified, and stored as encrypted information [6]. Rejeb et al. [7] mention that this potential makes this technology open to the new IoT-based era (Internet of Things), which can offer greater transparency, traceability, and security throughout data transfer. It gained popularity with the emergence of cryptocurrency and bitcoin due to its applicability in finance [8]. However, the authenticity features of this technology paved the way for wider use in other non-financial business areas [9]. Blockchain is essentially different from most of the most popular programming software architectures because it consists of four distinct features: decentralization, visibility, traceability, and smart execution [10].

Thus, this work has been organized as follows: Section 2 describes the typical structure of a measurement system of a production unit and addresses the main technological aspects of the blockchain. Section 3 details the research method considered. Section 4 describes the main components of the proposed solution. Section 5 summarizes the conclusions, limitations, and possible need for further studies.

The proposed solution focuses on the use of blockchain technology in the measurement chain, from the field sensors to the regulatory agencies, which are directly responsible

for the management of the concession contracts. Once this technology deals with public resources, security in measurement operations is critical because it is directly linked to the payment of royalties. Thus, blockchain is a viable alternative to promote robustness in this process.

## 2. Fiscal Measurement Systems and Blockchain

To introduce the proposed solution, there is a need to understand some aspects related to oil and gas measurement regulations, especially concerning the concepts of fiscal measurement and custody transfer. The comprehension of such aspects also makes it possible to understand how blockchain technology can be an important driver to improve those processes.

### 2.1. Fiscal Measurement and Custody Transfer Aspects

Fiscal measurement and custody transfer are often confusing terms in the market. Custody transfer is more related to the purchase and sale of products, through which both parties agree to meet specific rules for the oil quantification. These laws deal with matters such as inconstancy, repeatability, linearity, and metrological proof (such as calibration of measuring equipment), established in contracts. Fiscal measurement is a broader concept encompassing the volumetric quantification of product mass for taxation, sharing, and royalty purposes [11]. Commonly, fiscal measurement bases itself on technical regulations issued by regulatory bodies of concession or sharing contracts, which refer to standards published by official or even private entities. Typically, these regulations set the guidelines to help the field operators to ensure complete and accurate results [12]. For example, they outline aspects including but not limited to the assessment of pressure and temperature, uncorrected fluid flow and volume measurement, calculation algorithms for gas and oil, uncertainty estimation, and instrument and meter calibration. In a broader analysis, it is possible to conclude that both the transfer of custody and the fiscal measurement ensure that the quantification of products, crude oil, or natural gas, is as correct as possible and count with proper traceability. It is necessary to guarantee the transactions between buyers and sellers (in the case of custody transfer) or between governments and concessionaires (in respect of fiscal measurement). Both serve the same good practices applied to the quantification of these products. It is also noteworthy that, while fiscal measurement is more restricted to the fulfillment of concession contracts, custody transfer has much more extensive applications, including carriers and distributors of gaseous and liquid hydrocarbons, as well as their derivatives, reaching the final consumer.

The calculation approach for the value of the product, considering the requirements of fiscal measurement or custody transfer, is divided into large groups of equipment: (a) the measuring station installed in the field and (b) the supervision station, which interfaces with the various field-measuring stations.

In Figure 1, we have a diagram of a measuring station to evaluate the amount of natural gas meeting the requirements of fiscal measurement using an orifice plate as a primary element (flow calculation). In this case, we can consider that it consists of sensors, which measure the primary variables such as pressure (identified as PT in the diagram), temperature (TT), and differential pressure (PDT); the plate, which is the generating element of the differential pressure proportional to the gas flow (FT), and the flow computer (FQI), which runs an algorithm from the primary variables and flow parameterizations [13]. The flow computer is the "recording box" for the results of the measurement, from which one can obtain the normalized flow value of the gas at a certain pressure and base temperature, as defined in the technical regulations [14].

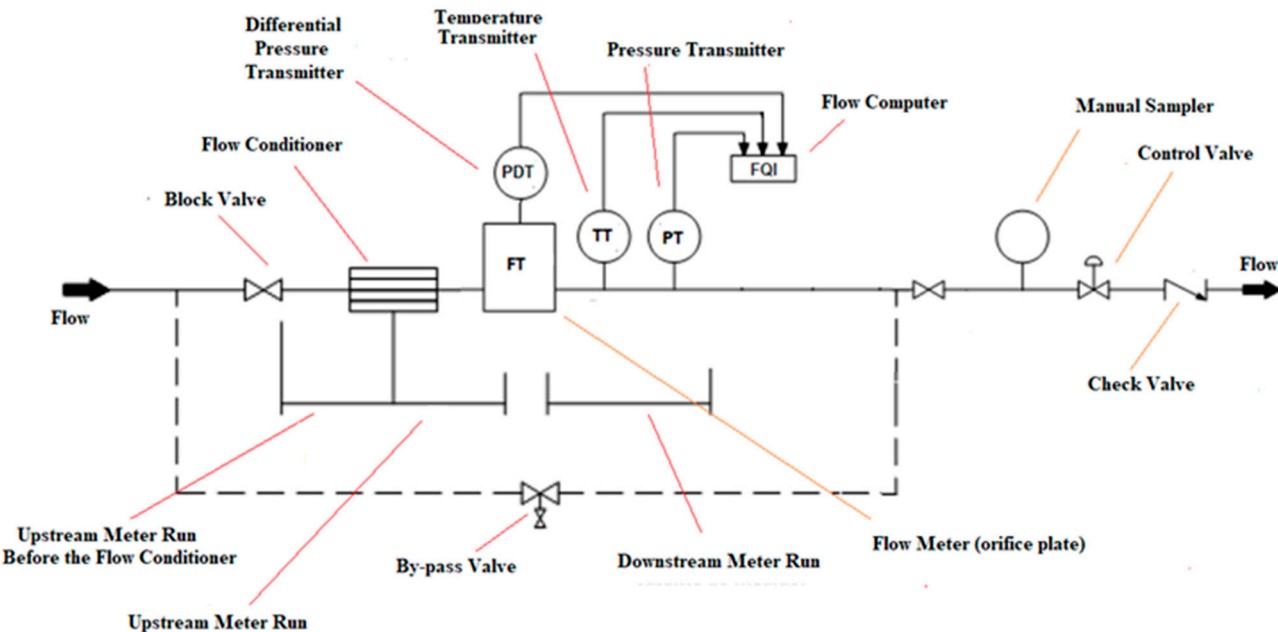

**Figure 1.** Fiscal Measuring Station Scheme for Natural Gas [14].

It is in the flow computer where events history and alarm databases, which are used in quantification, are generated [15]. The historical databases contain the average values of the primary variables, the main flow correction factors, and the parameters used in the calculations [16,17]. These databases are protected, and the final consumer cannot modify the information.

Figure 2 introduces the components of the Supervisory Station, which communicates with the various Measuring Stations installed in the production unit. This station plays an important role in the tax measurement system, which is the collection of databases generated by the flow computers. The flow computers generate these databases, but there is a limit to the storage capacity in its memory. Typically, it can store data of the last 250 alarms, 250 events, and 35 days of hourly average values [18], with the data stack operating on the First-In-First-Out system; in other words, the Supervisory Station needs to collect the databases before the information is lost [19]. Naturally, this process can be done manually without the use of a Monitoring Station, but in this case, operators would need to travel to the field to directly collect this information from each flow computer, which might be impractical in most installations.

The Supervisory Station is responsible for another important role: generating the reports on the measurement systems. Among the reports generated, there is a particularly important one: the Monthly Report. It is the Production Bulletin that provides information for royalty assessment and production profit sharing. In the case of royalties, to calculate the basic value of the production, it is necessary to consider the yielded volumes (also called exported), the volumes consumed in the production unit (largely for energy generation), the burned-off volumes (mainly in the gas flare), and the deducted volumes that come from other units (called imported). Similarly, the same analysis should be conducted for profit-sharing contracts.

In Figure 3, we have a typical schematic model of an offshore production system where we can identify the fiscal measurement points (identified with the letter F), the allocation measurement points (identified with the letter A and serving as a distribution of fiscal measurement across production wells), and the operational measuring points (identified by letter O).

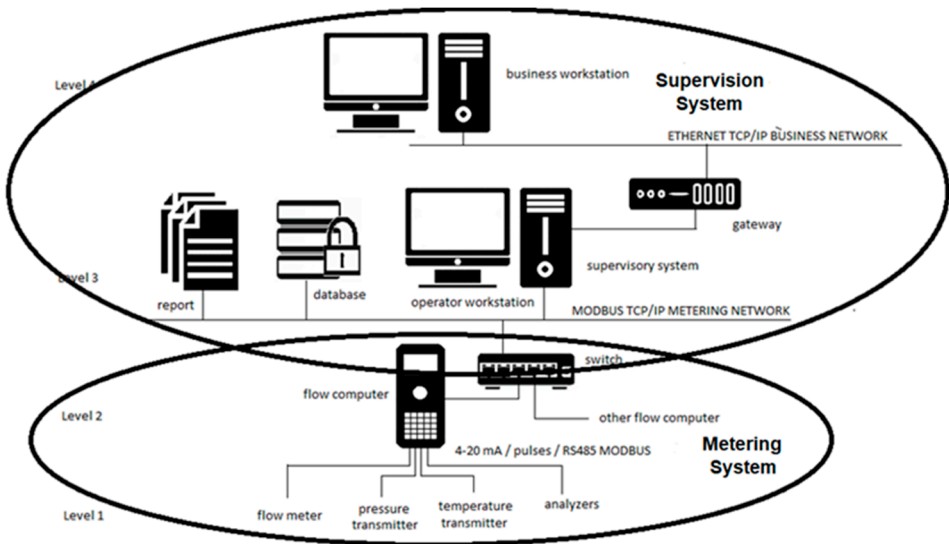

**Figure 2.** Supervision Station Scheme for Fiscal Measurement System [19].

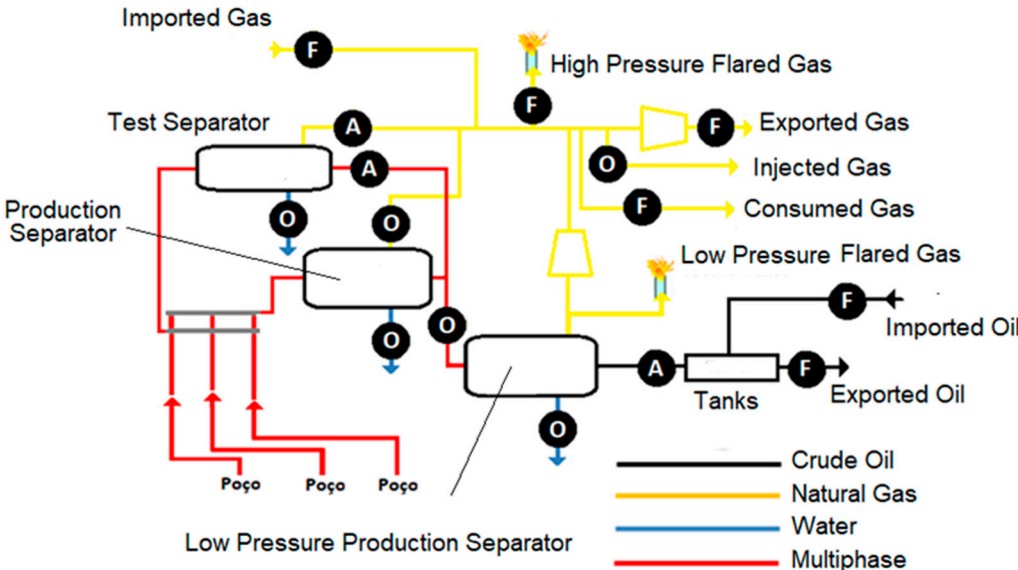

**Figure 3.** Fiscal Measurement Points in an Oil and Gas Production Unit.

The great weakness of this system, however, is that the Measurement Bulletins have no protection regarding data inviolability. Additionally, in many places, they are even generated manually. Similarly, in many cases, the Supervisory Station also lacks a safety system. Despite this fact, there is a certain degree of data protection between the flow computers and the field sensors, and all third-party interventions, which may affect the measurements, are recorded. However, above from the computers (as summarized in the diagram), the data are completely vulnerable.

### 2.2. Blockchain Technology

Blockchain is a technology that ensures the security of transactions with crypto assets, as it allows tracking of the sending and receiving of information between parties. In each transaction, blocks with a specific mathematical function aggregate with each other, which can generate an alphanumeric sequence. In addition to the transaction code itself, each block also has the code from the previous one, allowing them to connect and ensure that there was no infringement. The connection between the different blocks forms a data

stream, hence the name blockchain [20]. Blocks are added linearly and chronologically to the blockchain.

Finding the blocks, unraveling the codes, and joining them to other blocks is a very complex task. Therefore, those who execute it (miners) receive a reward for their work. In other words, mining is nothing more than the validation of the blocks. A signature (or "hash") is added to the end of each block.

As the blocks are added in sequence, the information is saved according to the processing time. Thus, after their validation, the blocks are recorded in a book (ledger), which contains all the information. Anyone can access them, but the sender's identity and address are protected. No one can delete or alter the data recorded. The network is in the cloud, which means that it is not centralized anywhere. In addition, there are several layers of security, eliminating the possibility of hacking. If there is such an attempt, the system automatically crashes.

According to Kshetri [21], blockchain emerges as a technology that can provide a solid and robust cyber security solution and a high level of privacy protection. The advocates of blockchain argue that such technology is safe by design. In a blockchain-based model, there is no need to store information with third parties.

For many authors, part of the fascinating character of blockchain derives from the fact that private forum data must be viewed and accessed only with the permission of the real owner, and such information cannot be stored. Proof of identity [22], as such a process is called, is stored in a cryptographic format, making it impossible or very difficult to compromise [23].

The security features of many of the essential systems in many industries depend on the so-called "security through obscurity" approach [21] in security engineering. Studies in this field aim to keep the security mechanisms and the implementation of the system in complete anonymity. Although the primary disadvantage of this method is that the entire system can collapse when someone finds out the security mechanism, according to numerous research, such practice becomes practically impossible with the innovative technology of blockchain.

Information privacy has a variety of definitions, and Bélanger and Crossler [24] provide a review of multiple interpretations. They conclude that a common theme among most of them is the control over personal information, especially concerning secondary use of this data by third parties. The definition of transparency is not as clear as the definition of privacy. Ball [25] identifies three metaphors related to transparency: "transparency as a public value embraced by society to combat corruption, transparency as a synonym of open decision-making by governments and non-profit organizations, and transparency as a complex tool for good governance in programs, policies, organizations, and nations".

Blockchain technology can guarantee all these three aspects by using encrypted keys. Noticeably, blockchain does not work as the traditional methods, and as the process for breaking the cryptography is extremely complicated, it also ends up being extremely safe [26].

The terms privacy and transparency seem contradictory at first glance. Blockchain design allows it to be an open technology, through which anonymous transactions are visible to anyone. Therefore, it ensures privacy and transparency examination side by side. At a high level, it is possible to distinguish between a public and a private blockchain. The literature mentions the terms "open" or "without permission" to refer to a public blockchain, and "with permission" to a private one [27,28].

The problem with privacy in blockchain technology is that all transactions happen openly and are identifiable by their to-be-validated hash value [29]. In a public blockchain, everyone has access to it; therefore, everyone can participate unconditionally in the decision-making and validation process.

The idea of a private blockchain is to monitor recording permissions given by a central decision-making entity and to restrict or allow reading permissions of individual users [28]. The level of decentralization and anonymity also differentiates blockchain in public and

private. A private blockchain process can never reach the same level of decentralization as a public blockchain, having implications for the degree of transparency.

A public blockchain is completely transparent, while a private one has a degree of transparency that can be controlled. When it comes to blockchain transparency, it means that the transactions are visible, but the parties involved remain anonymous [30]. Privacy and transparency in blockchain, therefore, depend on the type of blockchain used.

Blockchain paves the way for a "trust-free" economy where no reliable third party is necessary due to its transparency and highly secure design [26,31]. Blockchain is a techno-social system, whose technical component ensures the transactions of the social sector [26]. Once in blockchain, there is not a single instance that depends on trust in third parties, therefore, this credibility should be created differently. This is achieved by having all participants mutually unreliable, and thus credibility is generated through the blockchain consensus mechanism [32,33].

In the blockchain, consensus algorithms are created to maintain network reliability. They guarantee that no data entered will be erased and that the new information in the blocks has the credibility of all and cannot be violated. There are some types of consensus algorithms, the following ones being the most common:

- Proof-of-work, or proof-of-work, which creates a competition between computers connected to the blockchain, stimulating the mining of new blocks.
- Proof-of-stake, or proof of participation, through which participants who own blockchain digital assets have rewards in block validation.

Although there are previous studies, blockchain technology was first described by Nakamoto [8], creator of bitcoin, in the document "A Peer-to-Peer Electronic Cash System", which served as the basis for the concept of cryptocurrency. In the material, he points out that blockchain forms a record that cannot be changed without a reset in all operation steps from the beginning. Although the origin is linked to the financial market, blockchain technology can also be used to validate documents (such as contracts) and all types of transactions, including buying, and selling assets.

The academy has focused on applications of blockchain technology and Table 1 summarizes the main published papers related to the operations' modeling, performance metrics, operations' digitization, operations' costing, payment system, and smart contracts, which are the focus of this work.

**Table 1.** Related Study.

| Tools and Methods | Authors |
|---|---|
| Operations' Modeling | Babbitt and Dietz [34]; Christidis and Devetsikiotis [29]; Burcher, Decker, and Wattenhofer [35]; Harer and Fill [36]; Rosu et al. [37]; Frank [38]; Kim and Laskowski [39]; Lankhorst [40]; Luu et al. [41]; Sape [42]; Van Eck et al. [43]; Farah et al. [44]; Jaipuria and Mahapatra [45]; Yang et al. [46]; Mao et al. [47]; Al-Saqaf and Seidler [48]; Wang et al. [49]; Kshetri [50]. |
| Performance Metrics | Nields and Moriz [51]; Groenfeldt [52]; Hannam [53]; Higgins et al. [54]; Swafford et al. [55]. |
| Digitization of Operations | Treiblmaier [56]; Douglas and Kandaswamy [57]; Lyons-white and Knight [58]; Bridges and Fowler [59]; Popper and Lohr [60]; Childerhouse et al. [61]; Tykhonov, Jonker, Meijer, and Veewaart [62]; Durach, Kurpjuweit, and Wagner [63]. |
| Operations' Costing | Zhang and Wen [64]; Zhang et al. [65]; Cheah and Fry [66]; Raisaro et al. [67]. |
| Payment System | Nanayakkara et al. [68]; Behera, Mohanty, and Prakash [69]; Moon, Abd-karin, and Danuri [70]; Abeysekara, Wang, and Kuruppuarachchi [71]; Khaqqi et al. [72]. |
| Smart Contracts | Zhang et al. [73]; Casey and Vigna [74]; Correia [75]; Frias [76]; Gomes [77]; Ioannis et al. [78]; Miles [79]; Orcutt [80]; Wright and De Filippi [81]. |

Blockchain technology today is already in its second generation, being identified by the term "Blockchain 2.0", which emerged in 2014 and was based on a new distributed database project. In other words, information can be scattered across multiple computers.

The Ethereum network could have been one of the outcomes that stemmed from the implementation of the second-generation blockchain [82].

Currently, this technology is no longer limited to financial matters. Blockchain is incorporated in the processing and manufacturing, healthcare, logistics, and supply industries. New applications are emerging, given the reliability in the quality of the information. The expectation for the use of blockchain in the future is to serve millions of users. To provide them with a more diverse set of services, the underlying technology must consider large-scale storage and assessment behind the scenes [83]. The use of detection techniques for a similar transaction that can mitigate the redundancy of transactions in a distributed ledger is equally important for systems' rationalization [84].

## 3. Methods

This paper aims to describe the solution developed for a tax-measurement system in an oil and natural gas production unit, using blockchain technology to ensure transparency, efficiency, and optimization. These aspects concern the data flow from the sensors and field meters, which run through the flow computers and the monitoring stations until it reaches the drawing up of the Monthly Report. This one in turn informs the regulator about the volumes produced, consumed, and injected into the reservoirs.

The proposed solution focuses on the use of blockchain technology in the measurement chain, from field to oil and gas regulatory agency, which is directly responsible for managing concession contracts. As we deal with transacting public resources, security in measurement operations is critical because it is directly linked to the payment of royalties. Thus, blockchain is a viable alternative to create robustness in this process.

The solution, developed within the university using the necessary knowledge in conjunction with the support of equipment manufacturers and production system operators, can be implemented to meet the measurement regulations of any country.

The implementation of blockchain technology must be carried out on the flow computer databases—the real cash register of the measurement system. There were no major difficulties in creating this functionality since modern equipment is already integrated with Internet of Things technologies.

## 4. Discussion

Concepts referred to as the Internet of Things (IoT) and machine-to-machine (M2M) communications have attracted a lot of publicity and interested groups, as well as many face-to-face meetings. IoT refers to the increasing connectivity of objects of all kinds—from home appliances to devices used in industrial applications, either to the Internet or to an Internet-like structure. The general idea behind this effort is that any smart device should be able to communicate with each other or with human interfaces anywhere on the planet. Hence, driving improvements in M2M-Industrial-productivity networks is an intelligent decision and widely deployed. One can benefit from IoT technology if done correctly but could also suffer if done with a lack of planning and caution [85].

The IoT has certainly come a long way since Ashton created this term as the title of a presentation at Procter & Gamble Company in 1999 [86]. Quite a lot has been written about IoT over the past years ever since, with many researchers and analysts correctly predicting an almost exponential increase in connected devices. The industry, as expected, has also been following this trend.

As the IoT has been evolving, the industry has also been developing new ways to tackle important issues like privacy, security, and how to deal with the enormous amount of data generated by connected devices. In addition to developing technology to address these issues is the added difficulty of developing solutions that fit the business needs of the served markets. The IoT crosses boundaries between enterprises, users, and consumers, further complicating the technical and business needs of solutions that will be accepted in the market. To differentiate the web-based efforts and call attention to industrial needs, Storey et al. [85] called this application the Industrial Internet of Things (I2oT).

For the world of automation, I2oT represents the opportunity for the convergence of industrial automation communication on a huge scale. It allows improvements in functionality, security, flexibility, ease of use, and cost savings. In the long run, it is good to increase operational efficiency. In the short run, it is a disruptive technology. Its adoption requires changes, consequently threatening entities that do not have the resources or leadership to implement the changes. The technology also crosses organizational borders because the challenges of noticing the benefits of this technology are naturally more organizational than technical [87].

The I2oT, cyber-physical systems, and cloud computing are the basis for the smart factory concept. For this, we need to (a) monitor physical processes, create a virtual copy of the physical world, and make decentralized decisions; (b) communicate and cooperate in real-time among M2M and humans; (c) have internal and cross-organizational services covering all the participants of the value chain [88].

In the future, it is feasible to believe that all devices that can benefit from an Internet connection will be connected. Internet of Things (IoT) technology is a key facilitator for this concept, delivering machine-to-machine (M2M) and machine-to-person communication on a massive scale.

The great advantage of IoT sensors is that they can provide transparency, trust, and collaboration, leading to a whole new set of data behaviors, although their security protocols have not yet been optimized and implemented. Online sensors cannot carry software encryption protocols due to energy saving and memory space requirements. Therefore, there is a predominant need for new architectures to keep costs down and, at the same time, preserve the efficiency of industries that use IoT technology. However, Internet connectivity can, in most cases, be associated with vulnerability to hackers, thus allowing data manipulation. In recent years, efforts have been made to standardize security protocols on IoT systems. Blockchain is one of the most suitable solutions once it deals with security issues and has the potential to become a standard to ensure security in a corporate data environment [89].

The integration of field sensors into the blockchain has undeniable advantages:

- Autonomous coordination: we can have a direct interaction between them and the demands of the final users.
- Point-to-Point Messages: Connected devices interact through a distributed ledger, exchanging data in a cryptographic environment to prevent any interception attack.
- Distributed file storage: The use of an encrypted public blockchain would take on the data storage from these sensors, allowing standardized cloud storage.
- Autocorrection: The immutable and distributed record of transactions created by blockchain combined with the interoperability of IoT devices opens opportunities for network autocorrection and regulation for greater efficiency and security.

However, there are two major problems in the full use of IoT in fiscal measurement, which are the focus of this work: (a) how sensors connect to flow computers and (b) how the flow computer algorithm operates. Even today, it is very common for pressure, temperature, and flow sensors to be interconnected to flow computers with analogue connections (standard 4–20 mA). There are relatively few systems through which those instruments and meters can connect to the flow computers digitally.

These few experiences are limited to the use of the HART protocol (Highway Addressable Remote Transducer), Modbus with TCP (Transmission Control Protocol), RTU (Remote Terminal Unit), and ASCII (American Standard Code for Information Interchange), Fieldbus Foundation, and Profibus PA [19]. However, even in cases of connection to digital communication, there would be local networks typically limited to 32 addresses. It could cause a great limitation once it prevents the occurrence of one single address for the device to use to build the blockchain.

Furthermore, the way the flow computers run the algorithms is another major constraint. Figure 4 depicts the diagram of this operation valid for orifice plate measurement technology. Every second, the electronic unit of the flow computer obtains the data of the

primary variables (pressure, temperature, and flow), calculates the average values of this acquisition, runs an algorithm, and gets the flow value at operating conditions, which is corrected for the specific mass of the fluid calculated for the reference conditions (based on pressure and temperature). Only after these actions does the flow computer express the corrected (normalized) instantaneous flow value. This one, by its turn, along with the values used in or obtained from the calculations, is stored in an internal database.

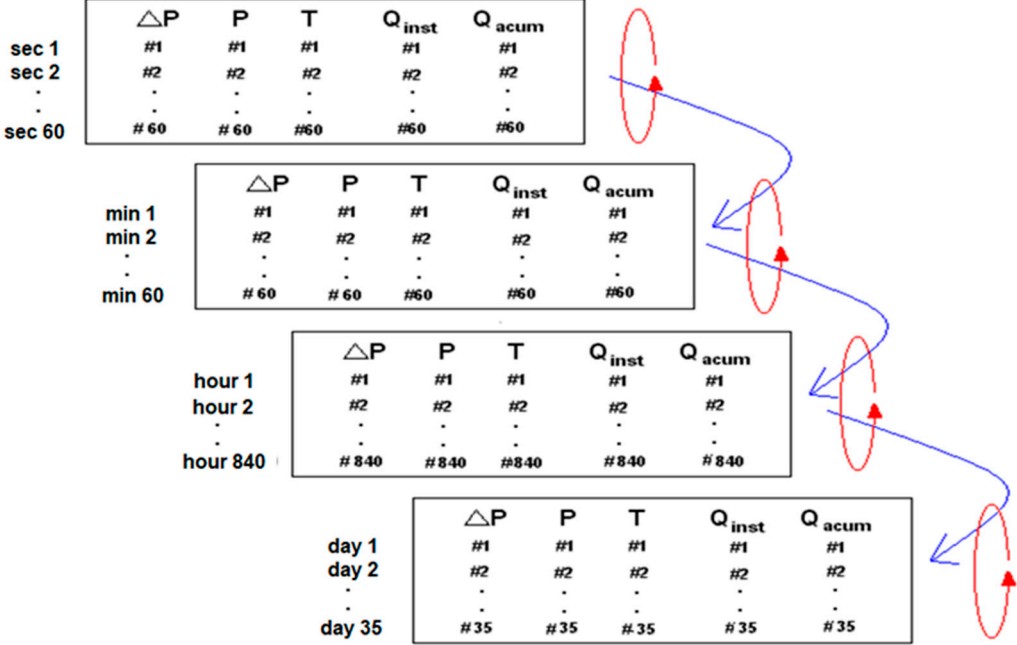

**Figure 4.** Construction of Historical Databases by Flow Computers.

The process is repeated after every minute, and the flow computer calculates the average values that are stored in another internal database. Successively, the process is repeated using the First-In-First-Out system, until the last-minute mean value is obtained. All these values are then stored in a database that can be accessed by the final user. This way, the flow computer keeps creating new lines continuously in the database every hour. It is observed, therefore, that there is no way to create an encryption that makes it possible to trace the data of the input variables, even because it would be raw data that would not have yet been processed by the flow computer algorithm [90].

Therefore, the blockchain solution needs to operate from the flow computer upstream of the measurement system architecture. In other words, the blockchain system must be deployed throughout the transmission of data from the flow computer to the Supervisory Station, and from the latter to the generation of the Monthly Report. This is possible because the typical communication between the flow computer and the Supervisory Station is executed through a TCP-IP or similar digital Ethernet network, which is the same system that supports communication between the Supervisory Station and the generation of the Monthly Report. In this case, both the flow computers and the Supervisory Station would have unique addresses through the MAC (Media Access Control address).

The MAC address is the most basic identification of a network card or device. Each card has a unique address, which is the base address of all network devices. There is only one MAC address for each board produced. When packages are sent or received, the devices compare the destinations and the adaptor's MAC addresses. If they are the same, the package is processed; otherwise, the information is discarded [91].

MAC is a 48-bit serial number whose function is to identify an Ethernet or Wi-Fi network device globally. The connections work with two protocols: the IP address and the MAC. The IP address is located one layer above the Ethernet, being the first option used in sending packets. If the destination is not in the same IP network as the sender, it is necessary

to send it to the configured router. Ethernet uses MAC addresses for these submissions. The ARP (Address Resolution Protocol) is used to recover the destination's MAC address. With the recovered address, the sender records the MAC address in the package and sends it to the recipient, where it is received and processed at its destination [92].

With the use of the MAC address, it is then possible to create the block of the blockchain and ensure data protection of the flow computer and the Supervisory Station itself, which in turn propagates this safety to the production report generator (Monthly Report). With the unique address of the information, we can then run the blockchain algorithm, which will add the transaction code to the block through its own algorithm. Put simply, the regulator would then have a Monthly Report protected by blockchain up to the level of the flow computer, as well as from the latter to the sensors.

Figure 5 summarizes how the proposed system would work for the incorporation of the blockchain algorithm in the information of the measurement of the production of the field.

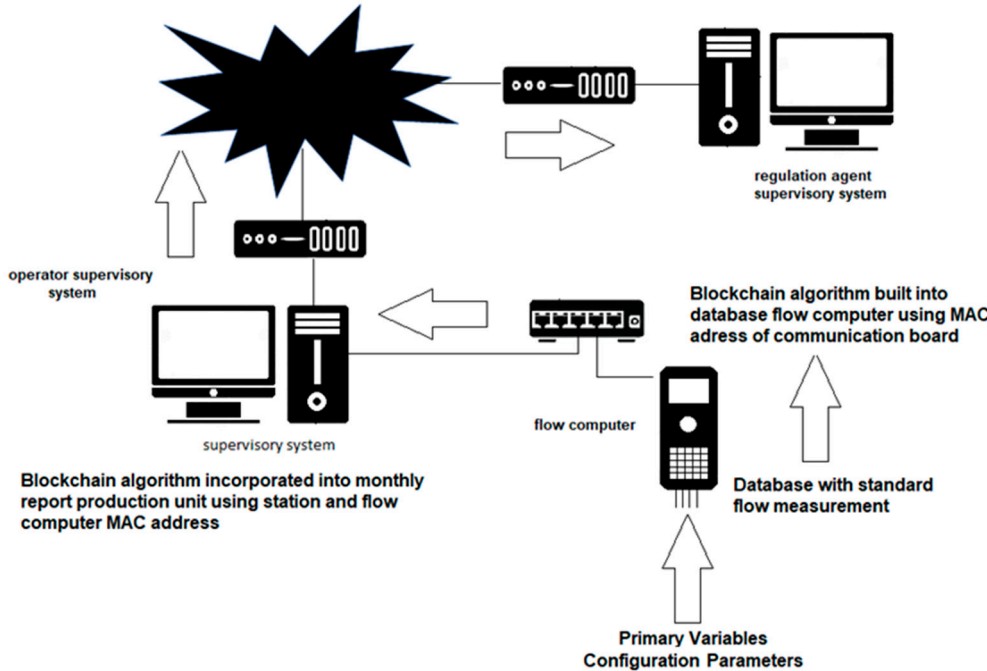

**Figure 5.** Incorporation of the blockchain algorithm in the measurement bulletin.

Evidently, using the MAC address is not a perfect solution. Many controllers and operating systems can change this address. Nonetheless, in oil and gas measurement systems, operators have a record, in their databases, of the data from the flow computers used in the calculation of product volumes (including the MAC address of the communication). Consequently, any change in the MAC address would make the received data incorrect. In other words, the use of the blockchain would have the main function of connecting the data received from the operator to its origin, the flow computer. This aspect alone justifies the use of the proposed solution.

## 5. Conclusions

The use of blockchain technology brings greater reliability in the generation of the Monthly Report, which is used for the calculation of royalties or even the oil and natural gas profit sharing—this one deriving from concession contracts. The Monthly Reports are the main result of the fiscal measurement process because they gather the information generated by the flow computers, the actual "cash registers" of the system. However, the generation of these documents is often hand-operated, hence subject to errors and failures.

With the adoption of the suggested proposal, the Supervisory Stations would then receive data from the flow computers, protected with the blockchain validation algorithm

that would, in turn, have the Monthly Report protected as well, once it would load the data generated by the flow computer. In other words, all data transfers would have more transparency, traceability, and security throughout the data transfer.

Ultimately, the proposed solution is based on the use of the MAC address (Media Access Control) as the primary input to the blockchain algorithm. To do so, it would be necessary that the communication between these devices take place with some digital protocol-type Ethernet TCP/IP, since the MAC addressing, which is unique globally, is only on the communication board of these devices.

As a suggestion for future studies, it would be recommended to make a more detailed implementation of the blockchain protocol in the flow computers in their output function to the Supervisory Stations, as well as in the communication between them and the field sensors, where extending blockchain technology to the lowest level of the system seems to be impracticable due to the way the flow computers calculate the normalized flow rate.

**Author Contributions:** Conceptualization, C.B., A.F., C.M. and J.F.F.; methodology, C.B., A.F., C.M. and J.F.F.; validation, J.F.F. and A.F.; formal analysis, C.B., A.F., C.M., K.M. and J.F.F.; investigation, A.F.; resources, C.B., A.F., C.M. and J.F.F.; data curation, C.B., A.F., C.M. and J.F.F.; writing—original draft preparation, C.B., A.F., C.M. and J.F.F.; writing—review and editing, C.B. and A.F.; visualization, C.B., A.F., C.M., K.M. and J.F.F.; supervision, J.F.F.; project administration, C.B.; funding acquisition, C.B., A.F., C.M. and J.F.F. All authors have read and agreed to the published version of the manuscript.

**Funding:** This research received no external funding.

**Conflicts of Interest:** The authors declare no conflict of interest.

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
