# Peer review of "Fiscal Measurement and Oil and Gas Production Market: Increasing Reliability Using Blockchain Technology"

_applsci, doi:10.3390/app12157874_

Round 1
Reviewer 1 Report
The topic of trust management became popular with the entry of DLT technology into maturity. The generic scheme is DLT, and the blockchain is one of its manifestations. The proposed work aims at the application of the blockchain in the delivery of a utility product. Resolves the basic issue of independence of control over the correct relationship between supplier and customer. The authors have aligned their research with the existing regulatory framework for oil supply control. The concept is based on the capabilities of the blockchain to manage trust between the parties in the supply chain.
I believe that their expectations for the security and protection of such a scheme are overestimated. At the very least, the MAC address is a specific category in terms of cybersecurity. Modern operating systems allow for changing the MAC address and even assigning more than one MAC address to one network controller. This does not detract from the concept in the study as it is a matter of specific technical implementation. I believe that the article should be accepted because it addresses the main question posed.
Author Response
Reviewer 1
- The topic of trust management became popular with the entry of DLT technology into maturity. The generic scheme is DLT, and the blockchain is one of its manifestations. The proposed work aims at the application of the blockchain in the delivery of a utility product. Resolves the basic issue of independence of control over the correct relationship between supplier and customer. The authors have aligned their research with the existing regulatory framework for oil supply control. The concept is based on the capabilities of the blockchain to manage trust between the parties in the supply chain.
I believe that their expectations for the security and protection of such a scheme are overestimated. At the very least, the MAC address is a specific category in terms of cybersecurity. Modern operating systems allow for changing the MAC address and even assigning more than one MAC address to one network controller. This does not detract from the concept in the study as it is a matter of specific technical implementation. I believe that the article should be accepted because it addresses the main question posed.
Our Comment: Of course, using the MAC address is not a perfect solution. Many controllers and operating systems can change this address, however, in oil and gas measuring systems, operators have a record, in their databases, of the data from the flow computers used in the totalization of volumes (including the MAC address of the of communication). So any change in the MAC address would make the received data unfeasible. In other words, the use of the blockchain would have the main function of connecting the data received at the operator with its origin (flow computer). This aspect alone justifies the use of the proposed solution. This text was incorporated into the paper manuscript. Revised the grammar and typing of the text.
Reviewer 2 Report
- Authors should add motivation and contribution in the introduction section
- Author should make one table in related study with existing work and their limitations?
- Authors should add some more specific areas where blockchain can be applied like industry 5.0, healthcare 5.0, and so on.
- All the table must be improved and the text within the table must be aligned properly.
- The grammar and typos error must be taken care
- Author should add advantages and disadvantages of the proposed model.
- Author should add some recent work (2021) in related study and should add a paragraph about scalability of blockchain applications. Kindly refer these papers to describe about scalability and duplicity of ledger content.
- Large-scale data storage scheme in blockchain ledger using ipfs and nosql. In Large-Scale Data Streaming, Processing, and Blockchain Security(pp. 91-116). IGI Global.
- Content-Based Transaction Access From Distributed Ledger of Blockchain Using Average Hash Technique. In Opportunities and Challenges for Blockchain Technology in Autonomous Vehicles(pp. 34-50). IGI Global.
Author Response
Reviewer 2
- Authors should add motivation and contribution in the introduction section
Our Comment: Added this paragraph In Introduction to clarify the importance and motivation of the paper.
“The proposed solution focuses on the use of blockchain technology in the measurement chain - from field sensors to the regulatory agent, which is directly responsible for managing concession contracts. As we are transacting public resources, security in measurement operations is critical because it is directly linked to the payment of royalties. Thus, blockchain is a viable alternative to create robustness in this process”.
- Author should make one table in related study with existing work and their limitations.
Our Comment: added to table 1 with the studies related to correspondence to the object of the work with the appropriate subjects and themes studied by the authors.
- Authors should add some more specific areas where blockchain can be applied like industry 5.0, healthcare 5.0, and so on.
Our Comment: the paragraph below has been placed to respond to the comment.
“Currently, this technology is no longer limited to financial matters. Blockchain is embedded in the processing and manufacturing, healthcare, logistics and supply industries. New applications are developed given the reliability in the quality of the information”.
- All the table must be improved and the text within the table must be aligned properly.
Our Comment: Tables and figures aligned with the margins.
- The grammar and typos error must be taken care
Our Comment: Revised the grammar and typing of the text.
- Author should add advantages and disadvantages of the proposed model.
Our Comment: Obviously using the MAC address is not a perfect solution. Many controllers and operating systems can change this address, however, in oil and gas measurement systems, operators have a record, in their databases, of the data from the flow computers used in the totalization of volumes (including the MAC address of the of communication). So any change in the MAC address would make the received data unfeasible. In other words, the use of the blockchain would have the main function of connecting the data received at the operator with its origin (flow computer). This aspect alone justifies the use of the proposed solution. This text was incorporated into the paper manuscript. Revised the grammar and typing of the text.
- Author should add some recent work (2021) in related study and should add a paragraph about scalability of blockchain applications. Kindly refer these papers to describe about scalability and duplicity of ledger content.
- Large-scale data storage scheme in blockchain ledger using ipfs and nosql. In Large-Scale Data Streaming, Processing, and Blockchain Security(pp. 91-116). IGI Global.
- Content-Based Transaction Access From Distributed Ledger of Blockchain Using Average Hash Technique. In Opportunities and Challenges for Blockchain Technology in Autonomous Vehicles(pp. 34-50). IGI Global.
Our Comment: Citation of the mentioned articles has been made.

Reviewer 3 Report
- The paper mainly discusses the impact of blockchain on increasing financial measurement and reliability in the oil and gas production market, but there are few practical approaches. The third chapter discusses the method, but separates into a paragraph is relatively abrupt. As the focus of this paper, this paragraph should be more detailed.
- The IoT is a broader concept, and its construction will apply the corresponding characteristics of blockchain. However, this article mainly deals with the application of blockchain, so the introduction of the IoT should be appropriately reduced, or directly introduced to the part that is more relevant to the thesis. And if it is the introduction of related concept or theoretical basis, it should be put in the second part.
- Line 364 says that “The integration of field sensors into the blockchain has undeniable advantages”, which is the focus of this paper and can be appropriately expanded. For example, what features of the blockchain are utilized by these advantages, and how these advantages affect financial measurement and the reliability of the oil and gas production market.
- The parameters in Figure 4 should be defined in advance.
- The blockchain solution (start at line 411) is the focus of this paper, which should be introduced in more space. Instead of just applying the characteristics of blockchain, it should reflect the specific aspects of reliability solved by the application of blockchain. It would be better to draw a flow chart to more clearly show the impact of blockchain application on reliability.
Author Response
Reviewer 3
- The paper mainly discusses the impact of blockchain on increasing financial measurement and reliability in the oil and gas production market, but there are few practical approaches. The third chapter discusses the method but separates into a paragraph is relatively abrupt. As the focus of this paper, this paragraph should be more detailed.
Our Comment: Chapter of “Methodology” revised.
- The IoT is a broader concept, and its construction will apply the corresponding characteristics of blockchain. However, this article mainly deals with the application of blockchain, so the introduction of the IoT should be appropriately reduced, or directly introduced to the part that is more relevant to the thesis. And if it is the introduction of related concept or theoretical basis, it should be put in the second part.
Our Comment: The concept of Internet of Things (IoT) introduced in the paper is important to address the issue of communication between the flow computers and the supervisory system, as well as between the flow computers and the field sensors. Modern communication protocols allow the incorporation of blockchain algorithms into the important data generated by flow computers. And once there are limitations regarding the proper functioning of the calculation method implemented by the regulations for the flow computers, it is not possible to extend the protection to the sensors. For this reason, it is necessary to discuss more thoroughly the aspects of IoT. We ask to keep the text as presented in the article.
- Line 364 says that “The integration of field sensors into the blockchain has undeniable advantages”, which is the focus of this paper and can be appropriately expanded. For example, what features of the blockchain are utilized by these advantages, and how these advantages affect financial measurement and the reliability of the oil and gas production market?
Our Comment: There are two major problems in the full use of IoT in fiscal metering, which are the focus of this work: a) how sensors connect to flow computers and b) how the flow computer algorithm operates. Even today, it is very common for pressure, temperature, and flow sensors to be interconnected to flow comput-ers with analogue connections (standard 4-20 mA). There are relatively few systems through which those instruments and meters can connect to the flow computers digitally. Anyway, the way the flow computers run the algorithms is another major constraint. Every second, the electronic unit of the flow computer obtains the data of the primary variables (pressure, temperature, and flow), calculates the average values of this acquisition, runs an algorithm, and gets the flow value at operating conditions, which is corrected for the specific mass of the fluid calculated for the reference conditions (based on pressure and temperature). Only after these actions does the flow computer express the corrected (normalized) instantaneous flow value. This one, by its turn, along with the values used in or obtained from the calculations, is stored in an internal database. Therefore, the blockchain could only be implemented from the generation of measurement data by the flow computer onwards, but from that stage on, the incorporation of the blockchain algorithm to this data promotes an improvement in the reliability in such a way that they are not violated. And as the royalties paid for the concession are directly linked to this measurement, blockchain technology brings robustness to the process. These aspects are already mentioned in the text right after the cited paragraph. We ask to keep the text as presented in the article.
- The parameters in Figure 4 should be defined in advance.
Our Comment: The parameters shown in figure 4 depend on the type of measurement technology used in the system. It reflects the operation of the flow computers from the primary data (pressure, temperature, and flow). However, depending on the measurement technology used, we may have other parameters. So, to make this clear, a new sentence was added, restricting figure 4 to orifice plate measurement technology.
- The blockchain solution (start at line 411) is the focus of this paper, which should be introduced in more space. Instead of just applying the characteristics of blockchain, it should reflect the specific aspects of reliability solved by the application of blockchain. It would be better to draw a flow chart to more clearly show the impact of blockchain application on reliability.
Our Comment: Figure 5 was incorporated, showing the working scheme of the proposed solution.

Round 2
Reviewer 2 Report
Paper can be accepted in the same format
Reviewer 3 Report
This paper has been reviewed carefully. So, I suggest that it is accepted.